# Intranasal Delivery of miR133b in a NEO100-Based Formulation Induces a Healing Response in Spinal Cord-Injured Mice

**DOI:** 10.3390/cells12060931

**Published:** 2023-03-18

**Authors:** Camelia A. Danilov, Thu Zan Thein, Stanley M. Tahara, Axel H. Schönthal, Thomas C. Chen

**Affiliations:** 1Department of Neurological Surgery, University of Southern California, Los Angeles, CA 90033, USA; cdanilov@usc.edu (C.A.D.);; 2Department of Molecular Microbiology and Immunology, University of Southern California, Los Angeles, CA 90033, USA

**Keywords:** spinal cord injury, scar tissue, grip strength meter, grasping task, microRNA-133b, NEO100

## Abstract

Despite important advances in the pre-clinical animal studies investigating the neuroinhibitory microenvironment at the injury site, traumatic injury to the spinal cord remains a major problem with no concrete response. Here, we examined whether (1) intranasal (IN) administration of miR133b/Ago2 can reach the injury site and achieve a therapeutic effect and (2) NEO100-based formulation of miR133b/Ago2 can improve effectiveness. 24 h after a cervical contusion, C57BL6 female mice received IN delivery of miR133b/Ago2 or miR133b/Ago2/NEO100 for 3 days, one dose per day. The pharmacokinetics of miR133b in the spinal cord lesion was determined by RT-qPCR. The role of IN delivery of miR133b on motor function was assessed by the grip strength meter (GSM) and hanging tasks. The activity of miR133b at the lesion site was established by immunostaining of fibronectin 1 (FN1), a miR133b target. We found that IN delivery of miR133b/Ago2 (1) reaches the lesion scar and co-administration of miR133b with NEO100 facilitated the cellular uptake; (2) enhanced the motor function and addition of NEO100 potentiated this effect and (3) targeted FN1 expression at the lesion scar. Our results suggest a high efficacy of IN delivery of miR133b/Ago2 to the injured spinal cord that translates to improved healing with NEO100 further potentiating this effect.

## 1. Introduction

Spinal cord injury (SCI) is a devastating condition that leads to long-term impairment of function; the higher the level of injury, the more severe the neurological damage. The most common cause of SCI is direct trauma to the cord itself, or from damage to the bones and soft tissue surrounding the spinal cord. There is an important window of opportunity after the primary injury to the spinal cord to alleviate secondary injury mechanisms that can exacerbate spinal cord damage at the lesion site. In a typical clinical scenario, patients receive the standard spine decompression and stabilization in order to prevent or limit the neurologic injury in the presence of unstable spine [1,2]. However, the prognosis remains poor and patients often still have residual partial or total paralysis, depending on the severity of injury. Considering the fact that a traumatic event of the spine most often results in spinal cord injuries, when it comes to the medical intervention, intranasal delivery could be considered a practical and non-invasive alternative to intravenous administration that can be used in emergency situations.

Irreversible neurological deficits and a highly inhibitory micro-environment for the neuronal growth at the lesion site are the hallmarks of the traumatic SCI. Despite numerous pre-clinical studies investigating the mechanisms of scar formation, the therapeutic approaches for current clinical use fail in part because of an insufficient understanding of the mechanisms behind lesion scar formation and the therapeutic time window for medical intervention following an injury. Traumatic injury of the spinal cord leads to the formation of a lesion scar that is characterized by two components: a glial scar made by reactive astrocytes in the perilesional area [3] and a fibrotic scar in the lesion core [4] the latter is considered a major impediment to axon regeneration [5]. Accumulated research data of the injured optic nerve [6], sciatic nerve [7] or spinal cord [8,9] reveales changes in molecules that are involved in extracellular matrix (ECM) composition at the lesion scar. In this light, our recent research in a pre-clinical model of cervical contusion injury established that intravenous (i.v.) delivery of microRNA133b (miR133b) mimic along with Argonaute 2 (Ago2), an endogenous binding partner, decreases the expression of collagen 1 type alpha 1 and Tenascin N in the lesion scar when delivered 24 h post-spinal cord injury leading to a positive functional recovery outcome in mice [10]. While this finding clearly indicates a beneficial effect of miR133b/Ago2 complex on improving spinal cord recovery, we envisioned a few challenges for the clinical trial in acute/emergent situations: (1) i.v. administration might be difficult for injured patients under any circumstances and (2) in vivo degradation of miR133b by cellular and extracellular nucleases leading to decreased therapeutic effect. To overcome these drawbacks, in this study, we investigated a strategy that combines the co-administration of miR133b/Ago2 with NEO100, a highly purified GMP-produced version of perillyl alcohol (POH) via nasal route within 24 h post-SCI.

Perillyl alcohol (POH) is a naturally occurring monoterpene, a non-nutritive dietary constituent found in essential oils of citrus fruits, peppermint, spearmint, cherries, and herbs with antioxidant, anti-inflammatory and anti-tumor activities [11,12]. Scientific studies demonstrated its beneficial role as a neuroprotector against ischemia-reperfusion injury in a middle cerebral artery occlusion (MCA) model of stroke in rats [13] and against amyloid beta (Aß)- neurotoxicity in a cellular model of Alzheimer’s disease in SY5Y neuroblastoma cells [14]. Traditionally, POH purity ranges from 85% to 96% and the impurities can be potentially inhibitory towards the desired therapeutic effect of the compound. Therefore, in this study, we used NEO100, a highly purified (≥99%) version of POH. Recent studies from our group using a rodent glioblastoma model reported that intranasal (IN) delivery of bortezomib (BZM), a proteasome inhibitor with a poor blood-brain-barrier (BBB) penetration, with NEO100, resulted in a better absorption of the drug in the brain and tumor tissue when compared to BMZ delivery alone [15].

The existence of nose-to-spinal cord pathways has been shown by radiolabel studies where intranasal delivery of insulin-like growth factor-I (^125^I-IGF-1) or interferon-beta (^125^I- IFN-1ß) reached the cervical spinal cord by passing the BBB in rats and monkeys [16,17]. In regards to miR administration via the intranasal route, recently, one study showed that IN delivery of miR155 antagomir alleviated acute seizures in a pentetrazol-induced acute seizure model in rats [18], while another study reported that IN administration of miR146a rescued cognitive impairments in APP/PS1 transgenic mouse, a model of Alzheimer’s disease [19]. Currently, the main routes for miRs administration following SCI include: intravenous, intrathecal and exosome/virus- mediated delivery [20], while miR delivery via the nasal route is poorly characterized and remains to be explored.

The overall goals of the present study were: (1) to investigate whether IN delivery of miR133b/Ago2 can achieve therapeutic effect and (2) to determine whether NEO100-based formulation of miR133b/Ago2 can improve on this effect. In this study, we used a mouse model of moderate bilateral contusive type of injury at cervical 5th (C5) levels, which resembles the contusive injury in humans, where the dura remains intact [9]. This injury model produces bilateral functional forelimb deficits that are assessed by the grip strength meter and hanging behavior tasks.

Here, we first evaluated whether intranasal administration of miR133b/Ago2 contributes to improved functional recovery in injured mice when delivered 24 h post-injury. Secondly, we investigated whether administration of miR133/Ago2 in a NEO100-based formulation via nasal route results in a better cellular uptake and stability at the lesion scar when compared to miR133/Ago2 alone. Third, we determined whether this higher miR133b uptake in the presence of NEO100 is also translated in an enhanced healing in injured mice.

## 2. Methods

### 2.1. Experimental Design

All animal procedures were approved by the Institutional Animal Care and Use Committee (IACUC) of the University of Southern California, Los Angeles. A total of seventy-five C57BL6 female mice between 6 and 8 weeks of age and a weight of 20–25 g at the beginning of the experiment were used in this study. Mice were randomly divided into the following groups: (1) pharmacokinetic studies after a single intranasal delivery (n = 51; 3–4 mice per group); (2) behavior studies and histological analysis (n = 24 total; 3–5 mice per group). A comprehensive description of the spinal cord injury conditions and lesion types is presented in Table 1.

### 2.2. Spinal Cord Surgical Procedures

The spinal cord injury model used in this study was a bilateral contusion injury model at C5th level, as previously described [21]. Mice were deeply anesthetized through intraperitoneal injection of a cocktail of ketamine (100 mg/kg) and xylazine (10 mg/kg), and a dorsal laminectomy was performed on C5. The contusion injury was created using the Infinite Horizon (IH) device platform IH-0400 (Precision Systems and Instrumentation, LLC, Fairfax Station, VA, USA). To induce a bilateral injury, the impactor tip was positioned above the midline followed by a single impact. Between 3 to 5 days post-injury, mice received an injection of Ringer’s lactate solution (1 mL/20 g, SubQ) for hydration, buprenorphine-slow release (0.5 mg/kg, SubQ) for analgesia, and enrofloxacin (2.5 mg/kg, SubQ) for prophylactic treatment against urinary tract infection. 

### 2.3. Grip Strength Meter (GSM) Task

A 4-week handling and pre-training procedure was used prior to SCI during which the baselines were collected for all tasks. Testing was done blind with respect to treatment groups. Mice were trained on the GSM task using a device designed by TSE-Systems (Chesterfield, MO, USA), as previously described [22]. For testing, mice were held by the tail near t o the bar so that they reach out to grip the bar. To test grip strength of a single paw, the opposite forepaw was gently taped with non-stick surgery tape (Micropore^TM^ surgical tape from 3M, Cat# 1530-0) to prevent gripping. Grip strength of each paw was tested three times per week (10 trials/session). The mice were held facing the bar so that they did not reach at an angle during the trials. A positive grip was scored when the digits extended and then flexed upon contacting the bar, followed by the digits being extended as the mouse released the bar and landed on the platform. A score of zero was given when a clenched/closed forepaw engaged the bar or if the forepaw landed on the platform in a clenched/closed position. Mice were tested three times prior to injury and two times per week for 8 weeks post-injury.

### 2.4. Hanging Task

The hanging task is a simple approach to measure the ability of the forepaws to grasp and maintain grip by assessing the mouse’s ability to hang from a suspended metal rod as previously described [23] with few modifications. For testing, the GSM bar was placed vertically against the wall and raised to its maximum height to avoid mice leaning against the base. Hanging time was measured in seconds from the time a mouse was placed hanging by its forepaws on the bar until it fell. During the testing, hind paws were taped to prevent them from being used for climbing atop of the metal. Three trials were collected per session per mouse. The test was performed three times prior to injury and every two weeks for 8 weeks post-injury. The data are reported as hanging time in seconds.

### 2.5. Intranasal Administration

Mice received intranasal delivery of miR133b mimic (mmu-miR-133b-5p; Dharmacon, Cat# C-311305-00) with mouse Ago2 (Sino Biological Inc., Cat #50683-MO7B), or miR-Negative Control (a scrambled miR sequence) (miRNA Mimic Negative Control; Ambion, Cat # 4464058) together with Ago2 protein. The dose for each administration was 10 µmol/L (60 pmol/6 μL) of miRNA combined with 0.1 µmol/L (0.6 pmol/6 μL) of Ago2, dissolved in saline solution before administration in a total volume of 6 μL. For the NEO100-based therapy we used NEO100 compound (stock 6.3 moles/L), produced in compliance with current Good Manufacturing Practice (cGMP) regulations by Norac Pharma, Azusa, CA, USA and provided by NeOnc Technologies, Inc., Los Angeles, CA, USA. NEO100 represents a highly purified (99%) form of perillyl alcohol (POH). In this study NEO100 was dissolved in ethanol: glycerol (50:50) solution to reach the desired concentration. NEO100 concentrations used in this study were as follows: 12.6 mmol/L (0.2%), 63 mmol/L (1%) and 315 mmol/L (5%). IN administration was performed dropwise with a 10 μL pipette tip. The mice were held in a supine position and a total volume of 6 μL was slowly applied to alternating nostrils 3 μL/drop, one drop per nostril followed by 1 min waiting time. After the mouse had received all drops, the animal was kept restrained on its back until the material disappeared into the nares and then returned back to its cage. The treatment schedule was as follows: (a) For the detection of miR133b levels in the cervical spine and lesion, mice received a single dose of miR133b/Ago2 at 24 h post-SCI. (b) For behavioral studies, mice received 3 intranasal administrations at one dose per day for 3 consecutive days, beginning at 24 h post-injury as described in Figure 1.

### 2.6. Immunohistochemistry

Immunohistochemistry was performed as previously described [22]. Mice were euthanized with an overdose of ketamine and xylazine (160 and 20 mg/kg, respectively) and transcardially perfused with 4% paraformaldehyde (PFA) in 0.1 M sodium phosphate buffer (Na_2_HPO4, pH 7.4). Spinal cords were dissected and post-fixed in 4% PFA overnight, then immersed in 30% sucrose overnight for cryoprotection, frozen in TissueTek OCT (VWR International, Radnor, PA) and stored at –20 °C until sectioning with a cryostat. The tissue block extending from ~4 mm above to 4 mm below the lesion and containing the injury site was sectioned at 20 µm intervals in the horizontal plane. The sections were stained with specific antibodies for glial fibrillary acidic protein (GFAP; Dako, Cat# Z0334), fibronectin (FN1; Invitrogen, Cat# MA5-11981) and collagen 1 a1(Col1a1; Abcam, Cat # Ab21286). Free- floating tissue sections were collected in phosphate-buffered saline (PBS) every 5th section for each staining procedure. After blocking in 5% normal goat serum (a non-immune serum that contains enough endogenous proteins to saturate and block non-specific binding sites) in PBS solution, sections were incubated in primary antibody (GFAP 1:1000, FN1 1:500, Col1a1 1:200) at 4°C overnight in PBS with 5% normal goat serum. The following day, sections were washed in PBS and incubated with the fluorescent secondary antibody (anti-rabbit Alexa Fluor 488, Molecular Probes A-11034; anti-mouse Alexa Fluor 488 Abcam Ab 150117, 1:250) in PBS with 5% normal goat serum for 2 h at room temperature. After twice rinsing in PBS, sections were mounted on gelatin-subbed slides (0.5% gelatin and 0.05% chromium potassium sulfate; Sigma Aldrich, St. Louis, MO, USA) and air-dried. Cover slips were applied with VECTASHIELD, Antifade Mounting Medium with DAPI (Vector Laboratories, Burlingame, CA, USA; Cat# H-1200).

### 2.7. Quantification of Immunofluorescence

Images were obtained on a BZ-9000 (BIOREVO) fluorescence microscope (Keyence Corporation of America, Itasca, IL, USA). Fields for fluorescent imaging were selected at the lesion site while viewing under 10× magnification. For each sample, three non-overlapping images were acquired and stored as fluorescent images (.TIF format) for analysis. For the quantification purpose, we used 4–5 sections per mouse with n = 3–5 mice per group. To quantify the intensity of FN1 and Col1a1 staining, ImageJ software from NIH was used to compute the fluorescence intensity of these markers. Collected data were recorded and compared statistically. To quantify the lesion size, we measured the positive area of GFAP expression using the Adobe Photoshop software. The lesion area was estimated by two parameters: width (dorsal-ventral position) and height (the length from the rostral to the caudal position).

### 2.8. miRNA extraction and expression

MicroRNA was extracted from the tissue samples of the (a) C4–C6 spinal cord segments that contained the lesion; (b) prefrontal cortex (c) spleen and (d) liver using mirVana miRNA Isolation Kit (Invitrogen Cat # AM1561) and reverse transcribed using TaqMan MicroRNA Reverse Transcription Kit (Applied Biosystems), according to the manufacturers’ instructions. Gene expression was assessed by RT-qPCR using TaqMan Universal Master Mix II (Life Technologies), TaqMan assay for miR133b (Applied Biosystems Cat # 4440886), and TaqMan Control miRNA assay for U6 snRNA (Applied Biosystems Cat # 4427975).

### 2.9. Statistical Analysis 

Data from different independent experiments were analyzed using Prism software. The results were plotted as mean ± standard error of the mean. To determine the differences between two groups we used *t*-test; for comparison between three or more groups one−way Analysis of Variance (ANOVA) or two−way Repeated Measures (RM) ANOVA with Bonferroni’s, Tukey’s and Dunnett’s correction for multiple comparisons were employed. A *p*-value of less than 0.05 was considered statistically significant.

## 3. Results

We have previously demonstrated improved spinal cord recovery after intravenous administration of miR133b/Ago2 to mice that underwent a cervical contusion injury [10]. This encouraging finding motivated us to determine whether (a) miR133b/Ago2 administrated intranasally, would show the same benefit as delivered systemically and (b) we can facilitate miR133b cellular uptake, stability and eventual healing by using NEO100 as a miR carrier.

### 3.1. Intranasal Administration of miR133b/Ago2 Has a Positive Effect on Forelimb Gripping and Grasping Healing in Injured Mice

We first determined whether IN administration of miR133b/Ago2 would have a positive effect on motor function recovery in injured mice. 24 h after receiving the cervical bilateral injury, mice were administered one dose of miR133b/Ago2 per day via nasal route for 3 consecutive days. Figure 1 illustrates the research design for this study. The effect of IN delivery of miR133b on functional recovery was assessed by employing two behavior tasks: GSM for forelimb grip strength evaluation and hanging task for forelimb grasp assessment during an 8 week-testing time post-SCI. Because we used a bilateral spinal cord injury as our model, both left and right forepaws showed impaired motor functions. For the grip evaluation, the left and right forepaws were tested separately as described in Methods.

The gripping and landing procedures during GSM testing are presented in Figure 2Ai. Figure 2B, C shows the forelimb gripping for the left Figure 2B, and right paw Figure 2C, respectively. Mice in scrambled-miR group had a poor gripping performance with 1 out of 4 mice showing a modest recovery throughout the testing time. A spontaneous transient grip (left paw) was noticed on week 5 in another mouse, but this grip was not consistent in the following testing weeks. In contrast, mice receiving IN delivery of miR133b/Ago2 showed the first sign of recovery at 7 days post-administration and continued to improve up to 42 days post-injury (3 out of 4 mice) when they reached a plateau (two-way RM ANOVA, left paw: *p* = 0.036 and right paw: *p* = 0.026).

Representative images of the hanging task are presented in Figure 2D, Di. Figure 2E shows the forelimb grasping performance in both scrambled-miR and miR133b/Ago2 treated groups. During the first two weeks of testing, none of the mice in both groups were able to perform the task. As observed with the GSM task, the mice in the control group showed a modest performance throughout the study such as at the end of testing the average hanging time was 13 s when compared to miR133b/Ago2 group where the average time was 27 s. (*p* = 0.01, two-way RM ANOVA). Altogether, the behavioral data suggest that miR133b/Ago2 delivered intranasally has a positive effect on motor function recovery when compared to scrambled-miR group.

### 3.2. Intranasal Co-Administration of miR133b/Ago2 with NEO100 Facilitates the miR133b Cellular Uptake in the Contused Spinal Cord

The optimal NEO100 concentration that resulted in the highest miR133b accumulation at the injury site was determined by real time RT−PCR. 24 h post-SCI, mice received IN co-administration of miR133b/Ago2 with NEO100 at concentrations of 0.2% (12.6 mmol/L), 1% (63 mmol/L) and 5% (315 mmol/L). Another group received intranasal delivery of miR133b/Ago2 without NEO100 (miR alone group). For comparison, the amount of miR133b was determined in contused spinal cord in the absence of intranasally delivered exogenous miR133b mimic (injury only group). The pharmacokinetic studies were assessed at 30-, 90-, and 240-min post-administration as presented in Figure 3.

Figure 3A–C shows relative miR133b levels expressed as fold changes compared to levels measured in contused spinal cord isolated from mice without receiving intranasal miR133b delivery. IN administration of miR133b/Ago2 alone resulted in a 36.4-fold increase at 30 min followed by a 47.6-fold change at 90 min (Figure 3B). Furthermore, we found that addition of NEO100 at 1% concentration facilitated the most miR133b accumulation at the lesion scar, as follows: at 30 min there was a 111-fold increase in comparison to 0.2% where the fold change was 44 or 5% concentration that resulted in a 42-fold change increase (Figure 3A). At 90 min, the miR133b uptake was slightly increased with both 0.2% and 5% NEO100 concentrations such as 64-fold change (0.2%) and 59-fold change (5%), respectively. 4 h later, the highest level of miR133b detected in the lesion scar was in the presence of 1% NEO100 (8-fold-increase) when compared to 0.2% (1.4-fold-changed), 5% (2-fold-increase) or miR133b/Ago2 alone (3.7-fold-increase) (Figure 3C). There was no significant difference between miR133b/Ago2 alone and miR133b/Ago2/vehicle group at all three time points (*p* = 0.98; *p* = 0.17 and *p* = 0.74, respectively). These data suggested that miR133b/Ago2 delivered intranasally alone can reach the contused spinal cord within 30 min post-administration, peaks at about 90 min and gets degraded to a 3.6-fold increase 4 h later. On the other hand, IN co-administration of miR133b/Ago2 with 1% NEO100 shifted the cellular uptake so then the highest detected level was at 30 min post-administration and 4 h later the level was 2 times more than the miR level detected after IN delivery with miR133b/Ago2 alone. Based on these results, the 1% NEO100 concentration was subsequently used in our further studies.

### 3.3. Distribution of the Intranasally Delivered miR133b/Ago with and without NEO100 in Off-targets

Having determined the level of miR133b accumulated at the lesion scar following IN delivery, we further pursued the biodistribution in off-targets such as prefrontal cortex, spleen and liver. To determine the extent to which miR133b/Ago2 reaches these organs, the comparisons were made between the miR133b/Ago2 alone and miR133b/Ago2 co-delivered with 1% NEO100 groups. The levels of miR133b in off-targets were quantified as a fold change compared to the levels measured in prefrontal cortex, liver and spleen isolated from mice that received injury only without treatment.

Figure 4A–C shows relative miR133b levels in prefrontal cortex at different times post-delivery. IN administration of miR133b/Ago2 alone resulted in a 117.7-fold increase at 30 min, while addition of NEO100 at 1% concentration further increased the miR133b accumulation up to a 168-fold change (*p* = 0.06). Over the next 90 and 240 min, miR133b levels decreased rapidly in both groups and reached the levels of the endogenous miR133b existing in the injury only group (Figure 4B,C). In stark contrast with the level of miR133b detected in brain at 30 min post-administration, miR133b distribution in spleen and liver was rather very low. For example, at 30 min after the IN delivery, the level of miR133b in spleen increased 2.4-fold change in the miR133b/Ago2 alone group vs. a 1.5-fold change in miR133b/Ago2 co-delivered with 1% NEO100 (Figure 4D). Over the course of 4 h, miR133b levels further decreased and reached the levels of the endogenous miR133b present in the injury only group (Figure 4E,F). We also found that IN delivery of either miR133b/Ago2 alone or co-delivered with NEO100 did not result in an accumulation of miR133b in the liver at 30 min post-administration but rather the levels were significantly lower than the levels in injury only group (Figure 4G). At 240 min, there was an increasing trend in miR133b level in both miR133b/Ago2 alone and miR133b/Ago2 co-delivered with NEO100 such as: 3.14-fold change vs. 3.07-fold-change, respectively of these groups when compared to the injury only group (Figure 4I). Taken together, these data suggested that (a) miR133b/Ago2 delivered intranasally reached the brain within 30 min post-administration and the addition of 1% NEO100 further increased the accumulation of miR133b in the brain; (b) the level of miR133b delivered intranasally showed a reduced distribution in off-targets such as spleen and liver; and (c) addition of 1% NEO100 did not significantly affect the distribution of miR133b when compared with miR133b/Ago2 alone group.

### 3.4. Intranasal Co-Administration of miR133b/Ago2 with NEO100 Further Potentiates the Forelimb Grip Strength but Does Not Change the Grasping Performance

Elevated miR133b concentration in the contused spinal cord after IN co-administration of miR133b/Ago2 with 1% NEO100 prompted us to investigate whether this increased uptake is also translated into a better functional recovery of injured mice.

As shown in Figure 5A, B, spinal cord injury resulted in complete loss of grip strength in both the left and right paws in all groups. Co-administration of NEO100 with scrambled-miR showed no substantial improvement in forelimb gripping with the exception of one mouse that presented a grip strength more pronounced with the right paw, perhaps due to a slightly skewed injury to the right side (Figure 5B). In stark contrast, the mice receiving IN delivery of miR133b/Ago2 with 1% NEO100 showed a much stronger grip at week 2, which further improved to nearly 100% by the end of the testing period. A direct comparison between groups receiving either miR133b/Ago2 alone or miR133b/Ago2 with 1% NEO100 at the end of study (8 weeks) is presented in Table 2. The data show that there is an increasing trend in the percentage of recovery of the grip strength in both left (39% increase) and right paw (31.6% increase) in miR133b/Ago2 co-administered with NEO100 (gripping force: 65.13 (g) left paw and 66.98 (g) right paw) when compared to the miR133b/Ago2 alone group (46.84 left paw and 50.9 right paw). These data suggested that NEO100 might facilitate the accumulation and stability of the miR133b at the lesion scar that translates to better healing. Neither NEO100 alone nor vehicle treated groups showed substantial improvement when compared to miR133b/Ago2/NEO100 group.

Figure 5C presents the forelimb grasping ability in groups receiving IN co-administration of either scrambled-miR or miR133b/Ago2 with 1% NEO100. Addition of NEO100 significantly improved the hanging time in the miR133b/Ago2 treated group when compared to either vehicle or NEO100 alone mice (*p* < 0.05, two−way RM ANOVA). As observed with the GSM task, neither vehicle nor NEO100 alone had an effect on grasping recovery. Based on these results we can conclude that addition of miR133b/Ago2 improved the hanging time, but not NEO100.

### 3.5. Neither miR133b nor NEO100 Causes Physiological Changes over Time

To determine whether miR133b/Ago2 delivered intranasally or addition of NEO100 affected the overall health, mice were checked weekly for any signs of distress such as: grooming and hair coat, scratching, examination of motor postures (hunching or cowering in the corner of the cage), changes in body weight. Except for the changes in weight loss described below, none of the listed signs were noticed in mice during the testing time. As shown in Figure 6, injury itself caused a weight loss in the first two days post-injury in all the groups followed by the recovery period when the mice began weight regain. Groups receiving the IN co-delivery with NEO100 showed an extended weight loss for a week. Among these groups, mice treated with miR133b/Ago2 co-delivered with NEO100 showed a significant weight loss when compared to the miR133b/Ago2 alone group (*p* < 0.05) but recovered to the same extent when compared with other groups. At the end of the study the weight loss among all groups was in the same range as follows: scrambled-miR alone: −9.56 ± 3.38 SEM; miR133b/Ago2 alone: −10.64 ± 1.4 SEM; scrambled-miR/NEO100: −9.17 ± 2.8 SEM; miR133b/Ago2/NEO100: −10.07 ± 2.98 SEM. The pattern of weight loss in the vehicle and NEO100 alone groups were mostly the same as in the NEO100 co-treated groups: vehicle −7.2 ± 2.29 SEM; and NEO100 alone: −9.58 ± 2.48 SEM, respectively.

### 3.6. Intranasal Administration of miR133b/Ago2 with and without NEO100 Decreases Levels of Fibronectin and Collagen 1 Type 1 at the Lesion Site

We next investigated the functional activity of miR133b following an IN administration. At the end of behavior testing (56 days), mice were perfused and spinal cord tissue including the lesion scar was dissected and analyzed by immunohistochemistry for the presence of fibronectin1 (FN1) protein, a miR133b target.

As shown in Figure 7A, there was prominent immunoreactivity for FN1 protein within the cervical lesion site of mice that had received IN delivery of scrambled-miR post-SCI. In comparison, the FN1 immunoreactivity was significantly less in injured mice that had received IN delivery of miR133b/Ago2 suggesting that miR133b delivery via the nasal route reached the lesion scar and was functionally active based on levels of FN1 (see Figure 7Ai for quantification of these effects). Furthermore, addition of NEO100 to the miR133b/Ago2 group did preserve the miR133b activity as we detected a decrease in FN1 fluorescence intensity (Figure 7B). In contrast, in mice receiving IN administration of vehicle or NEO100 alone there was a strong FN1 protein expression (Figure 7Bi), suggesting that miR133b/Ago2 delivered intranasally reached the lesion scar to act on its target, FN1mRNA, reducing its expression level.

Based on our previous studies [10] that ECM molecules such as collagen 1 type 1 (Col1a1) and tenascin N (TNN) at the lesion scar were less abundant by immunoreactivity upon i.v. delivery of miR133/Ago2, we further investigated the level of Col1a1 that changed upon intranasal delivery of miR133b. As shown in Figure 8A, mice that received miR133b delivered via nasal route showed a decreasing trend, approximately 16%, in Col1a1 expression level when compared to the control treated group (Figure 8Ai, *t*-test, *p* = 0.23). Co-administration of miR133b/Ago2 with NEO100 further decreased the Col1a1 immunoreactivity at the lesion site up to 27% when compared to scrambled-miR/NEO100 group and 22% when compared to vehicle treated group (*p* < 0.05) (Figure 8Bi), suggesting that addition of NEO100 facilitated the accumulation of miR133b at the lesion scar leading to an increased effect of miR133b when compared to the miR 133b/Ago2 alone treated group.

Pearson correlation coefficient analysis revealed that there was a moderate negative correlation between FN1 expression and gripping recovery (r = −0.4) and a strong negative correlation between Col1a1 and gripping performance (r = −0.6) at 56DPI, in mice that received miR133b/NEO100 treatment. In contrast, we found a moderate negative correlation between FN1 and gripping (r = −0.2) and almost no correlation between Col1a1 and gripping (r = 0.1) when mice were treated with miR133b/Ago2 alone via nasal route (Appendix A). Altogether, the data suggested that addition of NEO100 facilitated the miR133b accumulation leading to a better miR133b function than the IN delivery of miR133b/Ago2 alone.

### 3.7. Intranasal Administration of miR133b/Ago2 with and without NEO100 and the Lesion Size

Finally, to confirm the injuries and evaluate the potential scar-reducing capability of IN administration of miR133b/Ago2 with and without NEO100, we determined the size of the lesions at 56 days post-SCI by measuring the expansion of immunoreactivity generated by glial fibrillary acidic protein (GFAP, a marker for astrogliosis and lesion identification) in horizontal sections of the spinal cord (Figure 9A,B). The dorso-ventral (D/V) and rostro-caudal (R/C) dimensions are presented in Figure 9Ai,Bi,Bii. Our analyses showed that in the miR133b/Ago2 only treated group (Figure 9A), there was a 15% decreasing trend in lesion height and 18% reduction in lesion width when compared to scrambled-miR group (Figure 9Ai) (*p* = 0.34, One-way ANOVA). We also found that co-administration of miR133b/Ago2 with NEO100 decreased the D/V extension when compared to either lesions of scrambled-miR133b/Ago2/NEO100 or vehicle/NEO100 only groups the lesion (Figure 9Bi). In contrast, the lesion height (R/C) was similar among treatment groups except in the scrambled-miR /NEO100 group which had a decreased size (Figure 9Bii). A direct comparison between IN delivery of miR133b/Ago2 alone and miR133b/Ago2/NEO100, revealed that addition of NEO100 decreased the lesion height by 15.8%, while the lesion width was slightly increased by 5%. All together, these data suggested that IN delivery of either miR133b/Ago2 alone in a combination with NEO100 has the potential of delaying or preventing the full extent of regular scar formation.

## 4. Discussion

Animal research over the past decade has shown that microRNAs (miRs) are important players in different neurological processes associated with spinal cord injury and resulting neuroinflammation and apoptosis [24,25,26,27,28,29]. The involvement of miRs in physiological and pathological events following a trauma of the central nervous system is beginning to receive increased investigative attention. There are a few studies in the literature that address the role of miRs in astrogliosis after SCI. For example, one study reported that mice transfected with adenoviruses that contained either miR-17 inhibitor or PTEN cDNA via intraspinal injection immediately after receiving a contusion injury at the T11 level, showed reduced glial scar formation and improved hind limb function [30]. Another group demonstrated that astrocyte-specific overexpression of miR-145 by intraspinal injection of lenti-gfap-miR-145 at the end of T9/T10 injury reduced the astrocytic density at the lesion border within the spinal cord, along with less cell proliferation and migration [31]. Moreover, a study by Sabin et al. demonstrated that miR-200 inhibition lead to differential regulation of genes involved in reactive gliosis, glial scar formation, extracellular matrix remodeling, and axon guidance in a model of spinal cord ablation in axolotls [32].

The function of miRs in fibrotic scar formation is less explored. However, it has been shown that administration of antagomiR-21-5p intrathecally for 3 days regulated functional recovery in a rat model of thoracic compression through a mechanism that targeted TGF-beta regulated fibrogenic activation of spinal fibroblasts [33]. Recently, our group reported that intravenous administration of miR133b with Ago2, when performed 24 h after cervical contusion injury, efficiently interferes with fibrotic scar formation by regulating a number of ECM molecules in injured mice [10].

The main routes for miR administration following SCI include: intravenous [10], intrathecal [34,35], intraspinal [30,31], and via exosome-mediated delivery [36]. Therapeutic intervention by supplying miRs via the nasal route following spinal cord injury is not well characterized and in need of further investigation. Only very few studies have explored the nasal-spinal cord route as a potential delivery pathway for potential SCI therapeutics. In this context, a study conducted in rats with complete SCI shows that intranasal administration of mesenchymal stem cell-derived exosomes (MSC-Exo) loaded with phosphatase and tensin homolog (PTEN) siRNA decreases the expression of PTEN, leading to enhanced axonal growth, neovascularization and functional recovery [37]. Another study demonstrated that IN administration of nerve growth factor (NGF) bypasses the blood brain barrier and enriches NGF levels and expression of NGF receptors in intact and injured spinal cord. Moreover, these data indicated that IN-NGF is able to generate an improvement of motor behavior in rats with a hemi-sectional injury at the T10 level [38].

The animal research presented in this current study provides a framework for understanding the IN-delivery mechanism of microRNAs alone and in a NEO100-based formulation. Our results establish its efficiency and point to the possibility that in the future this principle of administration might be applicable to clinical emergency situations following a neurotrauma event such as spinal cord injury. 

### 4.1. Rationale for the Approach

Our recent research in a pre-clinical model of cervical contusion injury established that intravenous (i.v.) delivery of miR133b mimic along with Argonaute 2 (Ago2), an endogenous binding partner, targets the microenvironment of the contused spinal cord and it can enhance the healing process when used within 24 h of injury in mice [10]. Although this outcome clearly indicates a beneficial effect of miR133b/Ago2 complex on improving spinal cord recovery, we envisioned a few challenges for the clinical application in acute/emergent situations: (1) i.v. delivery to the injured patients might be difficult in a timely fashion; and (2) in vivo degradation of miR133b by cellular and extracellular nucleases leading to decreased miR133b delivery to the injury site. Therefore, we developed this strategy of IN delivery of miR133b/Ago2 in a NEO100 based formulation to enhance the miR133b availability at the lesion scar. This combination therapy was tested in a mouse model of SCI such as moderate bilateral cervical contusion that resembles the contusive type of injuries in humans, where the dura remains intact.

### 4.2. Intranasal Administration of miR133b/Ago2 Reaches the Contused Spinal Cord and Addition of NEO100 Further Facilitates the miR Accumulation

Intranasal administration of miRs following a spinal cord injury has not been extensively studied and it remains a topic of discussion. In this study, we tested whether IN delivery of miR/Ago2: (a) reached the lesion scar; (b) was sufficiently stable and (c) the extent to which addition of NEO100 increased miR133b accumulation in the injured spinal cord. Our studies showed that miR133b/Ago2 delivered intranasally, 24 h post-SCI, can reach the injured cord within 30 min and it was detected up to 4 h later. We also demonstrated that IN co-administration of miR133b in a NEO100-based formulation resulted in a better miR lesion accumulation than delivery of miR133b/Ago2 alone. More importantly, addition of NEO100 increased the level of miR133b in the injured cord 3.7 times more than did miR alone. We also showed that IN delivery of miR133b/Ago2 led to a 36.4-fold increase at 30 min and co-administration of miR with 1% NEO100 further enhanced the level of miR133b up to 111-fold increase suggesting that IN delivery has a great potential of becoming the preferred way of delivering miRs to the brain and spinal cord in the emergency situations but not only, especially when co-administered with NEO100 as a vehicle. However, further pharmacokinetic studies are required to determine the mechanism of miR133b uptake in different cells present at the lesion scar. Previous studies from our group using a rodent glioblastoma model reported that IN delivery of bortezomib (BZM)—a proteasome inhibitor with poor BBB penetration—along with NEO100, resulted in better absorption of the drug in the brain and tumor tissue when compared to BMZ delivery in the absence of NEO100 [15]. Therefore, it appears that NEO100 might act as a carrier to enable direct nose-to-brain transport of otherwise BBB-impermeable agents. Recently, studies from our group have shown that IN co-delivery of miR18a and NEO100 improved the pharmacokinetic profile of miR18a without affecting its pharmacological properties in an Mgp^−/−^ mouse model of arteriovenous malformation (AVM) [39]. 

### 4.3. Intranasal Administration of miR133b/Ago2 Alone or with NEO100 Preserves miR133b Functional Activity

The functional activity of miR133b delivered intranasally post-SCI was tested by assessing the fibronectin (FN1) protein expression level in the injured spinal cord in mice. We previously reported that miR133b targets extracellular matrix (ECM) molecules such as Col1a1 and tenascin N (TNN) known to inhibit axon growth and healing [10]. Here, we found that IN delivery of miR133b/Ago2 alone had a modest effect on Col1a1 expression level when compared to the control group. However, addition of NEO100 significantly reduced the Col1a1 protein expression in comparison to either scrambled-miR/NEO100 or vehicle treated mice. Moreover, we found that mice receiving IN delivery of miR133b/Ago2 with NEO100 showed a stronger negative correlation with the gripping performance than the mice receiving miR133b/Ago2 alone. Among other ECM molecules, FN1 has been also documented to be upregulated following spinal cord injury and its high expression was correlated with poor recovery [24,25,26]. FN1 is a miR133b target (http://www.mirdb.org, accessed on 12 December 2021), with a target score of 85 and three seed sequences in the coding region. In this study, we found that FN1, was significantly reduced upon miR133b/Ago2 treatment when compared to the control group at 56 days post-SCI. miR133b can target both FN1 mRNA degradation or suppression of the FN1 protein synthesis based on the sequence complementarity between miR133b and FN1 mRNA target. In these studies, we have not investigated the mechanism by which FN1 expression decreases upon IN administration of miR133b/Ago2. Therefore, further studies are important to establish which pathway is involved in this decreased expression of FN1 at the lesion scar. Co-administration of miR133b/Ago2 in a NEO100-based formulation, showed that NEO100 did not affect the miR133b activity as we observed the same decrease in FN1 expression level when compared to miR133b/Ago2 alone treated group. We also found that in miR133b/Ago2/NEO100 treated mice there was a significant FN1 down-regulation when compared to control groups. Interestingly, we detected the same FN1 expression level with scrambled-miR, scrambled-miR/NEO100 and NEO100 alone suggesting that NEO100 does not change the inhibitory microenvironment at the lesion scar, and the effect that we have seen is solely related to the miR133b activity. How FN1 contributes to the inhibitory milieu in the fibrotic scar it is unknown.

Our histological analysis data revealed that among all the lesion types (Table 1), 54% were fluid filled cystic cavity type, while about 46% showed a fibrous type. The loss of fibrousness could be caused by miR133b since on average 80% of all lesions in miR133b/Ago2 treated groups showed fluid filled cavities. In contrast, we found that in scrambled groups 50% of the lesions showed loss of fibrousness, while in NEO100 and vehicle groups only 25% of the lesions.

### 4.4. Intranasal Administration of miR133b/Ago2 Promotes Forelimb Gripping Force Recovery and Addition of NEO100 Further Enhances the Functional Recovery

In this study we found that IN administration of miR133b/Ago2 post-SCI leads to improved healing in injured mice. The assessment of forelimb function was achieved by employing two different behavior tasks that measure grip and grasp recovery. The forelimb gripping was determined by two factors: (1) the grip strength as represented by the gripping force value and (2) the landing as this step requires digit extension upon landing. On the other hand, the hanging task determined the grasp recovery by measuring the time a mouse stayed suspended. This task also determined the balance and coordination. Performing different assessments provided us with more detailed insight into the functional improvements following the treatments. Here, we found that IN delivery of miR133b/Ago2 enhanced gripping and addition of NEO100 further facilitated the recovery in both paws with about 35% percent increase. During the testing period, we observed that mice receiving IN delivery of miR133b/Ago2/NEO100 showed less variability in their responses than the mice receiving miR133b/Ago2 alone. One possible explanation could be the increased accumulation of the miR133b in the presence of NEO100 at the lesion scar. Regarding the grasping recovery, the data showed the same extent of recovery in both miR133b/Ago2 alone and miR133b/Ago2/NEO100 treated mice. It is important to mention that improvements in one functional test (e.g., forelimb grip strength assessing sole muscle strength/flexor muscles) does not necessarily coincide with improvements in another test (hanging test assessing muscle function/balance/ sustained suspension).

## 5. Conclusions

The studies presented here demonstrated a high-efficacy of IN delivery of miR133b/Ago2 to the injured spinal cord and supported that IN delivery of miR133b/Ago2 in a NEO100-based formulation further enhanced miR133b accumulation at the lesion scar. To our knowledge, this is the first study investigating the principle of IN delivery of miRs in a mouse model of spinal cord injury. While the proposed studies are in young female mice, future studies can be designed to explore the impact of aging and sex in mediating the benefits of IN co-delivery of miR133b/Ago2/NEO100 in animal models. In addition, benefits of IN delivery of miR133b/Ago2 in a NEO100-based formulation seen in our SCI model may also show benefits in other models of neurodegenerative disorders. Our studies will begin to detail more specifically how miR is transported via the nasal route to the contused spinal cord and furthermore how NEO100 can be used as a general vehicle to deliver miRs.

## Figures and Tables

**Figure 1 cells-12-00931-f001:**
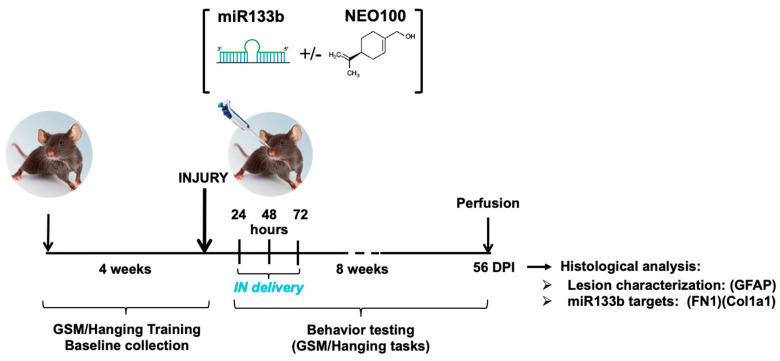
Schematic representation of the experimental design. Mice were trained for gripping and grasping tasks for 4 weeks followed by the baseline collection. At 24 h post-injury, mice received the intranasal delivery of either miR133b/Ago2 alone or miR133b/Ago2 with NEO100 for 3 consecutive days. The control groups received scrambled−miR alone with or without NEO100, NEO100 alone and Vehicle alone. The behavior testing was performed for 8 weeks post-SCI. At the end of all testing, mice were euthanized, spinal cords dissected and further analyzed for lesion identification (GFAP staining), fibronectin (FN1) and Collagen1 type 1 (Col1a1) protein expression. DPI = days post-injury.

**Figure 2 cells-12-00931-f002:**
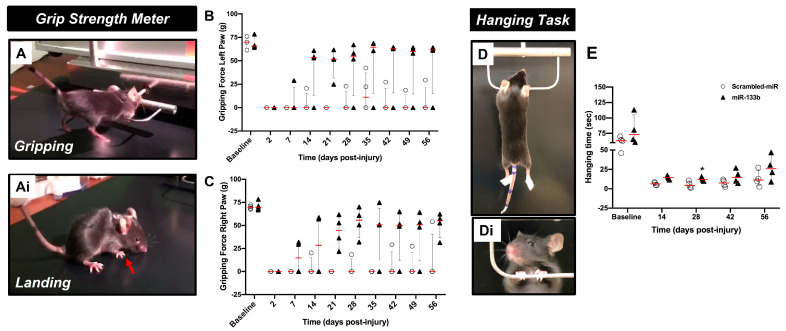
The effect of intranasal delivery of miR133b/Ago2 alone on forelimb gripping and grasping recovery in mice. The graphs represent the forelimb gripping (panels **A**–**C**) and grasping (panel **D**,**E**) assessments during 56 days of recovery, post-injury (DPI) in miR133b/Ago2 and scrambled−miR treated mice. Panels (**A**,**Ai**) show representative images of the gripping and landing procedures during the GSM testing. Panels (**D**,**Di**) represent the grasping procedure during hanging task. Two−way Repeated Measures, ANOVA, followed by the Bonferroni’s multiple comparison test. * *p* < 0.05 when compared miR133b to scrambled−miR group. The circles and triangles in each graph represent the individual mouse performance per group. The red line shows the median with interquartile range. The red arrow indicates the plantar position at landing.

**Figure 3 cells-12-00931-f003:**
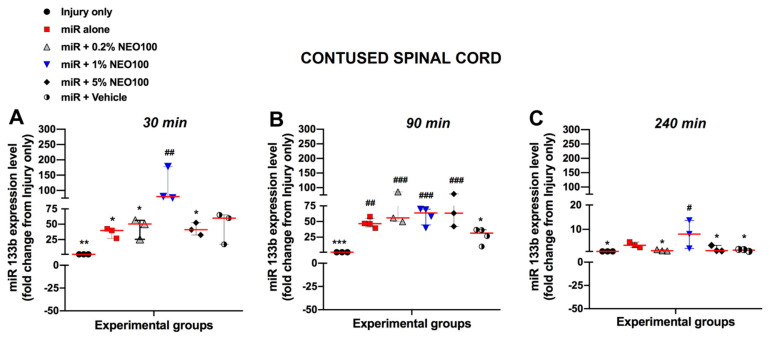
The level of miR133b in the contused spinal cord after a single dose of either miR133b/Ago2 alone or miR133b/Ago2/NEO100 administered intranasally. At 24 h after injury, mice were administered one dose of miR133b/Ago2 alone, or in combination with concentrations of NEO100 between 12.6 to 315 mmol/L, intranasally. Another group of mice received miR133b/Ago2 with vehicle (37% ethanol: glycerol) and the last group received injury only without intranasal treatment. After 30, 90 and 240 min, mice were euthanized and spinal cords that included the lesion scar were collected and processed for miRNA isolation. The level of miR133b was assessed by RT−qPCR. Panels (**A**–**C**) represent the level of miR133b at different time points expressed as a fold−change from injury only group. One−Way ANOVA with Dunnett’s multiple comparison test. ^#^*p* < 0.05 ^##^
*p* < 0.01; ^###^
*p* < 0.001 when compared with injury only group; *** *p* < 0.001 ** *p* < 0.01, * *p* < 0.05 when compared with miR133b/Ago2 + 1% NEO100 group. The circles, triangles and squares in each graph represent the individual mouse miR133b/Ago2 level per group. The red line shows the median with interquartile range.

**Figure 4 cells-12-00931-f004:**
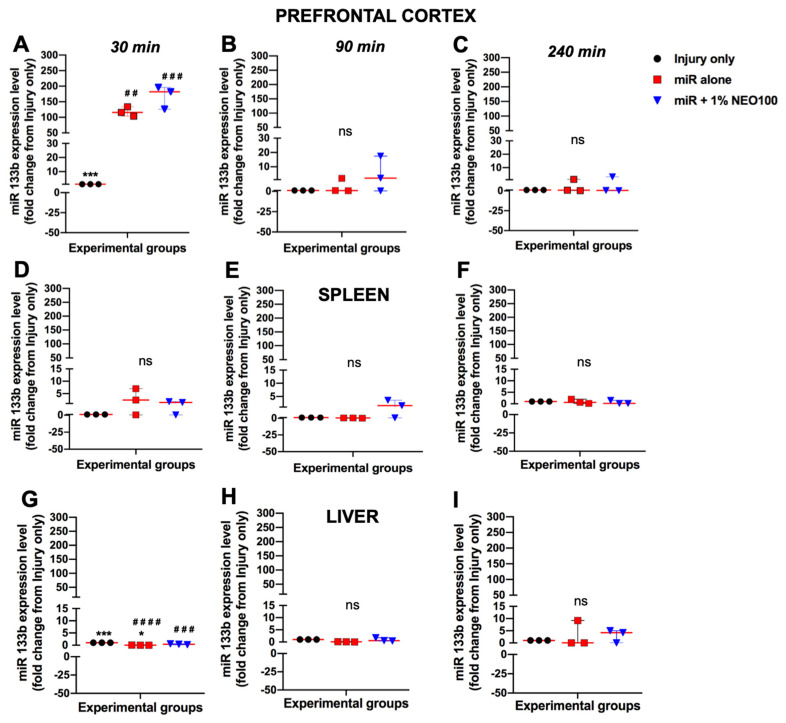
The level of miR133b in the prefrontal cortex, liver and spleen after a single dose of either miR133b/Ago2 or miR133b/Ago2/NEO100 administered intranasally. The prefrontal cortex, liver and spleen were collected from the mice receiving the treatment as presented in Figure 3. The level of miR133b in off targets is represented as follows: Panels (**A**–**C**) prefrontal cortex; panels (**D**–**F**) spleen and panels (**G**∓**I**) in liver. The data show the level of miR133b as a fold−change from mice that received injury without any treatment. One−way ANOVA with Dunnett’s multiple comparison test. ^##^
*p* < 0.01; ^###^
*p* < 0.001; ^####^
*p* < 0.0001 when compared with injury only group; *** *p* < 0.001; * *p* < 0.05 when compared with miR +1% NEO100 group. ns = the difference is not statistically significant among groups. The circles, triangles and squares in each graph represent the individual mouse miR133b level per group. The red line shows the median with interquartile range.

**Figure 5 cells-12-00931-f005:**
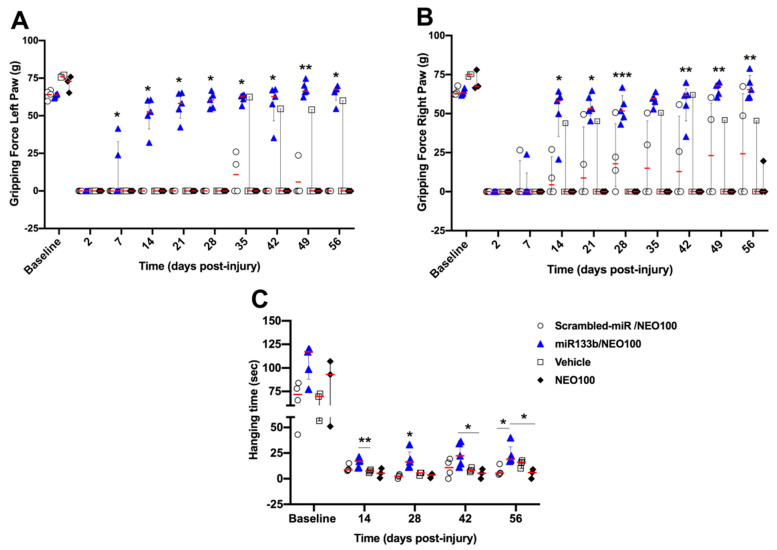
The effect of intranasal co−delivery of miR133b/Ago2 with 1%NEO100 on forelimb gripping and grasping recovery. The graphs represent the forelimb gripping (panels **A**,**B**) and grasping (panel **C**) assessments during 56 days post−injury (DPI) in miR133b/Ago2/NEO100, scrambled −miR/NEO100, Vehicle and NEO100 only treated mice. Two−way Repeated Measures, ANOVA, followed by the Tukey’s multiple comparison test. *** *p* < 0.001, ** *p* < 0.01, * *p* < 0.05. The circles and triangles in each graph represent the individual mouse performance per group. The red line shows the median with interquartile range.

**Figure 6 cells-12-00931-f006:**
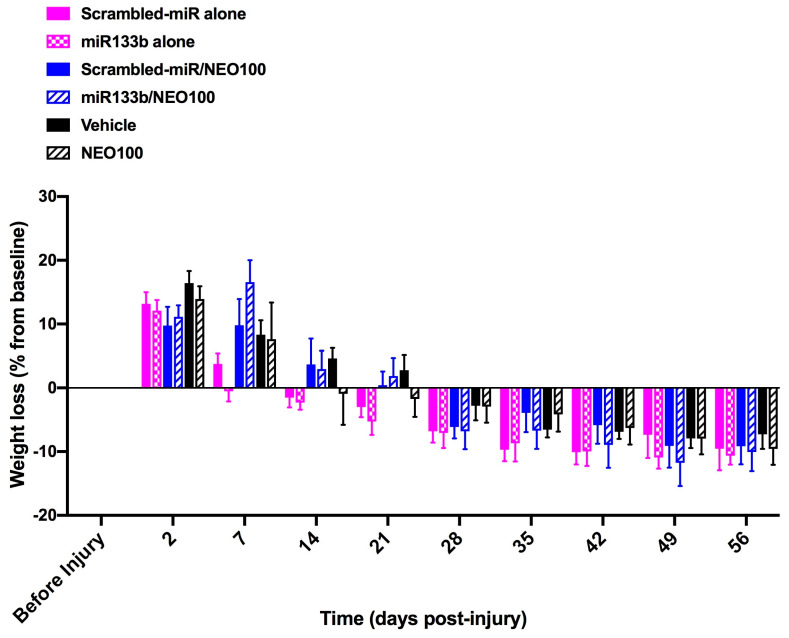
Weight loss assessment over the course of 56 days following the intranasal administration of either miR133/Ago2 alone or in combination with NEO100. The graph represents the weight loss over the course of 56 days following different treatments. The data are presented as percentage change from baseline (before injury). Two−way Repeated Measures ANOVA followed by the Tukey’s multiple comparison test.

**Figure 7 cells-12-00931-f007:**
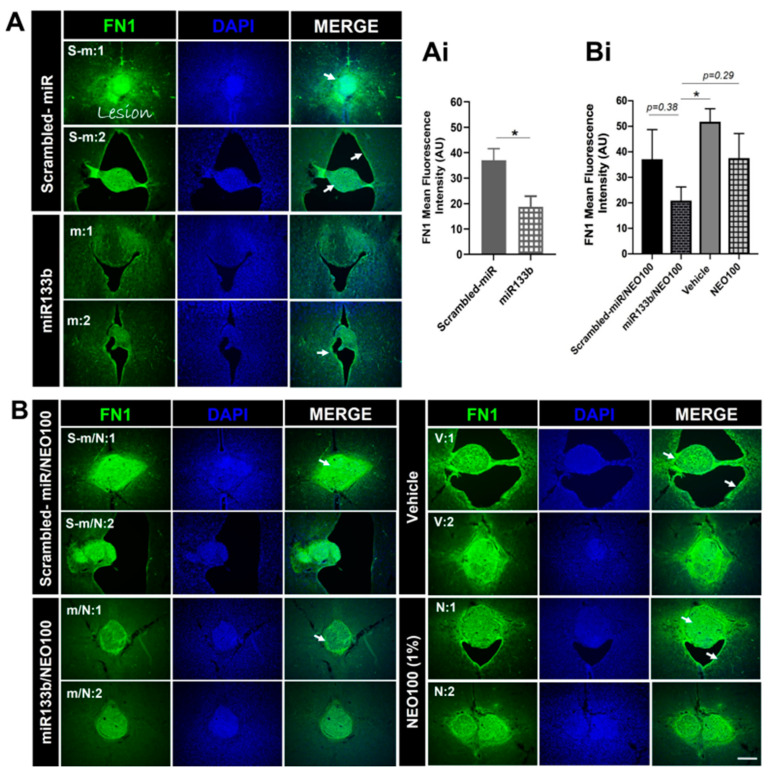
The effect of intranasal delivery of miR133b/Ago2 either alone or in combination with 1% NEO100 on the fibronectin (FN1) protein expression at the lesion scar at 56 days post-injury. Panels (**A**,**B**) show representative images of the spinal cord sections immunolabeled with FN1. Sections were collected from two different mice per group, as follows: s-m:1 and s-m:2 from scrambled−miR group; m:1 and m:2 from miR133b/Ago2 group; s−m/N:1 and s−m/N:2 from scrambled −miR/NEO100; m/N:1 and m/N:2 from miR133b/Ago2/NEO100 group; v:1 and v:2 from vehicle and N:1 and N:2 from NEO100 group. Panels (**Ai**,**Bi**) represent the fluorescence quantification of FN1. T-test was employed for comparing the groups in graph (**Ai**). One−way ANOVA with Dunnett’s multiple comparison test was used for comparing the groups in graph (**Bi**). * *p* < 0.05; n = 4–5 sections per mouse, n = 3–5 mice per group. Note: The arrows indicate the FN1 expression at the lesion scar. Scale bar: 100 μm.

**Figure 8 cells-12-00931-f008:**
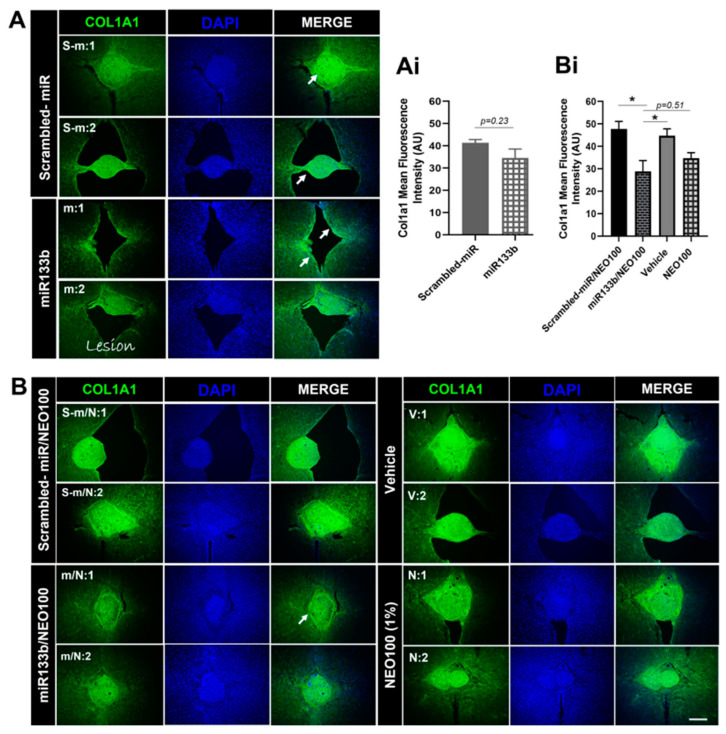
The effect of intranasal delivery of miR133b/Ago2 either alone or in combination with 1% NEO100 on the Collagen 1 type 1 (Col1a1) protein expression at the lesion scar at 56 days post-injury. Panels (**A**,**B**) show representative images of the spinal cord sections immunolabeled with Col1a1. Sections were collected from two different mice per group, as follows: s−m:1 and s−m:2 from scrambled-miR group; m:1 and m:2 from miR133b/Ago2 group; s−m/N:1 and s−m/N:2 from scrambled-miR/NEO100; m/N:1 and m/N:2 from miR133b/Ago2/NEO100 group; v:1 and v:2 from vehicle and N:1 and N:2 from NEO100 group. Panels (**Ai**,**Bi**) represent the fluorescence quantification of Col1a1. T−test was employed for comparing the groups in graph (**Ai**). One−way ANOVA with Dunnett’s multiple comparison test was used for comparing the groups in graph (**Bi**). * *p* < 0.05; n = 4–5 sections per mouse, n = 3–5 mice per group. Note: The arrows indicate the Col1a1 expression at the lesion scar. Scale bar: 100 μm.

**Figure 9 cells-12-00931-f009:**
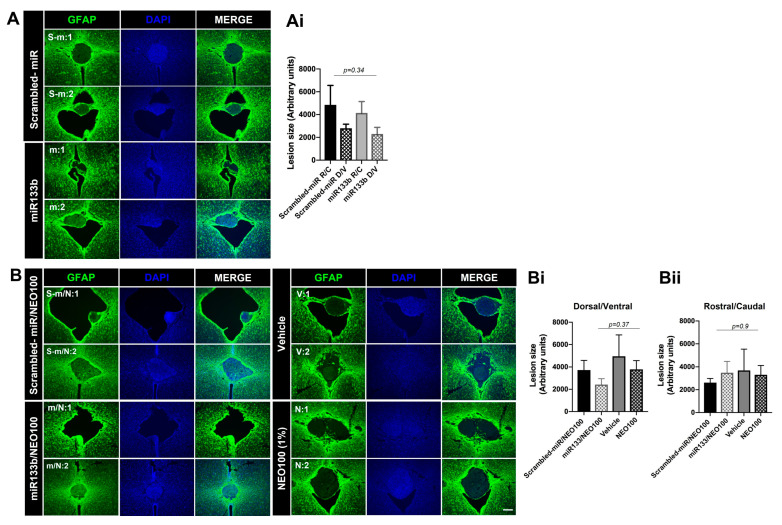
The effect of intranasal delivery of miR133b/Ago2 alone or in combination with 1% NEO100 on the lesion size at 56 days post-injury. Panels (**A**,**B**) show representative images of the spinal cord sections immunolabeled with GFAP for lesion identification. Sections were collected from two different mice per group, as follows: s−m:1 and s−m:2 from scrambled−miR group; m:1 and m:2 from miR133b/Ago2 group; s−m/N:1 and s−m/N:2 from scrambled−miR/NEO100; m/N:1 and m/N:2 from miR133b/Ago2/NEO100 group; v:1 and v:2 from vehicle and N:1 and N:2 from NEO100 group. Panels (**Ai**,**Bi**,**Bii**) represent the fluorescence quantification of the lesion size. One−way ANOVA with Tukey’s multiple comparison test. n = 4–5 sections per mouse, n = 3–5 mice per group. D/V = dorso-ventral (width) and R/C = rostro-caudal (height); scale bar: 100 μm.

**Table 1 cells-12-00931-t001:** A comprehensive description of the spinal cord injury conditions and lesion types for mice involved in behavior and histology analysis procedures.

Mouse #	Experimental Group	Type of Lesion	Force (KDyn)
#7-E12	Scrambled-miR alone	fibrous	81
#12-E12	Scrambled-miR alone	fibrous	83
#30-E13	Scrambled-miR alone	fluid filled cavity	84
#33-E13	Scrambled-miR alone	fluid-filled cavity	80
#9-E12	miR133b/Ago2 alone	fluid filled cavity	82
#10-E12	miR133b/Ago2 alone	fluid filled cavity	80
#14-E12	miR133b/Ago2 alone	fluid-filled cavity	82
#15-E12	miR133b/Ago2 alone	fluid-filled cavity	80
#19-E10	Scrambled-miR/NEO100	fibrous	83
#21-E10	Scrambled-miR/NEO100	fibrous	79
#14-E10	Scrambled-miR/NEO100	fluid-filled	83
#9-E10	Scrambled-miR/NEO100	fibrous/unilateral *	82
#10-E10	Scrambled-miR/NEO100	fluid-filled cavity	80
#7-E10	miR133b/Ago2/NEO100	fluid-filled cavity	85
#8-E10	miR133b/Ago2/NEO100	fibrous	79
#12-E10	miR133b/Ago2/NEO100	fluid-filled cavity	83
#15-E10	miR133b/Ago2/NEO100	fibrous	83
#22-E10	miR133b/Ago2/NEO100	fibrous/unilateral *	88
#20-E11	miR133b/Ago2/NEO100	fluid-filled cavity	86
#6-E12	NEO100 alone	fibrous	83
#11-E12	NEO100 alone	fibrous	81
#13-E12	NEO100 alone	fibrous	80
#8-E12	NEO100 alone	fluid-filled cavity	82
#37-E13	Vehicle alone	fibrous	81
*#34-E13*	Vehicle alone	Fibrous—incomplete *	81
#39-E13	Vehicle alone	fibrous	82
#36-E13	Vehicle alone	fluid-filled cavity	80

Note: Asterisks indicate the mice who were excluded from the behavior and histological analysis.

**Table 2 cells-12-00931-t002:** The comparison between gripping and grasping tasks in treated groups with and without NEO100 at the end of study (56 days post-SCI). Data are expressed as mean ± SEM.

Experimental Group	Gripping Force (g)		Grasping Time (s)
	Left Paw	Right Paw	
Scrambled-miR	7.34 ± 7.34 SEM ***	13.53 ± 13.5 SEM **	13.16 ± 5.19 SEM
Scrambled-miR/NEO100	0 ± 0 SEM ****	29 ± 17.18 SEM	7.25 ± 2.38 SEM
miR133b/Ago2	46.84 ± 15.63 SEM	50.9 ± 6.8 SEM	27 ± 8.04 SEM
miR133b/Ago2/NEO100	65.13 ± 2.7 SEM	66.98 ± 3.48 SEM	23.26 ± 4.28 SEM
Vehicle alone	20 ± 20 SEM **	15.16 ± 15.16 SEM *	14.56 ± 2.33 SEM
NEO100 alone	0 ± 0 SEM ****	6.54 ± 6.54 SEM **	5 ± 2.65 SEM*

One-Way ANOVA followed by the Dunnett’s for multiple comparison test. * *p* < 0.05, ** *p* < 0.01, *** *p* < 0.001, **** *p* < 0.0001 when compared to miR133b/Ago2/NEO100 group.

## Data Availability

The data presented in this study are available on request from the corresponding author.

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
