# Peer review of "Intranasal Delivery of miR133b in a NEO100-Based Formulation Induces a Healing Response in Spinal Cord-Injured Mice"

_cells, 2023, doi:10.3390/cells12060931_

Round 1
Reviewer 1 Report
I would like to congratulate the authors of the article, because it is very complete in terms of information and gives a detailed explanation of the methodology as well as the results. I believe that they meet all the objectives proposed in the project.
For this reason, I am accepting this article without corrections.
Author Response
Thank you very much for your thoughtful recommendation.
Reviewer 2 Report
This paper addresses the disease, SCI, without addressing its pathogenesis. The SCI initiates a very severe, destructive and very long inflammatory reaction that is counteracted by anti-inflammatory reaction mounted by the spinal cord. Any attempts at ignoring these pathologies relegate a study to a failure including conclusions. the BBB test is a poor indicator of changes occurring after the SCI, whether treated or not.
Author Response
Thank you for reviewing our manuscript.

Reviewer 3 Report
In the manuscript entitled “Intranasal delivery of miR133b in a NEO100-based formulation induces a healing response in spinal cord-injured mice” the authors investigate the effects of administering miR133b intra nasally with and without NEO100 formulation. This study is important and interesting in the field of spinal cord injury and neuroregeneration.
The manuscript is generally well-written.
I have a few minor corrections which might improve the manuscript
Typos: Throughout the manuscript space between words is not given, which may lead to a misnomer of the word.
The following statement in the Introduction section has mentioned miR133b/Ago2 “To overcome these drawbacks, in this study, we investigated a strategy that combines the co-administration of miR133b/Ago2 with NEO100, a highly purified GMP-produced version of perillyl alcohol (POH) via nasal route within 24 hours post-SCI” but in some places like simply mentioned as miR113b (for instance in abstract). Which one is correct?
Images can be added to the manuscript for Grip Strength Meter (GSM) task and Hanging task to increase the authenticity.
Table 1 is missing in the manuscript. It can be added to the manuscript / or as supplemental data.
The discussion is very weak in the manuscript and it has to be discussed with other studies to substantiate the results.
Author Response
Thank you so much for your thoughtful suggestions. Please see the attached revised manuscript that includes the revisions based on your suggestions.
Sincerely,
Camelia Danilov
